# Biological Activity and Component Analyses of *Chamaecyparis obtusa* Leaf Extract: Evaluation of Antiwrinkle and Cell Protection Effects in UVA-Irradiated Cells

**DOI:** 10.3390/medicina59040755

**Published:** 2023-04-13

**Authors:** Young-Ah Jang, Se-Gie Kim, Hye-Kyung Kim, Jin-Tae Lee

**Affiliations:** 1Convergence Research Center for Smart Healthcare of KS R & DB Foundation, Kyungsung University, Busan 48434, Republic of Korea; 2Department of Pharmaceutical Engineering, Daegu Catholic University, Gyeongsan-si 38430, Republic of Korea; 3College of Pharmacy, Kyungsung University, Busan 48434, Republic of Korea; 4Department of Cosmetic Science, Kyungsung University, Busan 48434, Republic of Korea

**Keywords:** flavonoid, *Chamaecyparis obtusa*, antioxidant, antiwrinkle, fibroblast, quercitrin

## Abstract

*Background and Objectives: Chamaecyparis obtusa (C. obtuse)* extract has been used as a folk medicinal remedy in East Asian countries to alleviate inflammation and prevent allergies. Active oxygen causes skin aging and leads to skin cell and tissue damage. Extensive research has been conducted to control active oxygen generation to prevent skin aging. We evaluated the antioxidant activity and antiwrinkle effect of *C. obtusa* extract to determine its potential as a cosmetic material. *Materials and Methods:* The antioxidant activity of a 70% ethanol extract of *C. obtusa* (COE 70) and a water extract of *C. obtusa* (COW) was determined using 2,2-diphenyl-1-picrylhydrazy (DPPH) scavenging, 2,2′-azino-bis(3-ethylbenzothiazoline-6-sulfonic acid) (ABTS^+^) scavenging, superoxide dismutase-like activity, xanthine oxidase inhibition, and ferric-reducing antioxidant power assays. The effective concentration of the extracts was determined using the methyl thiazolyl tetrazolium assay to evaluate their toxicity. The effects of COE 70 on the production of matrix metalloproteinases (MMPs) and procollagen, and expression of activated cytokines, interleukin 6 (IL-6) and tumor necrosis factor α (TNF-α), in UVA-irradiated fibroblasts were determined using quantitative real-time PCR. Additionally, quercitrin, amentoflavone, hinokiflavone, and myricetin concentrations in COE 70 were determined using high-pressure high-performance liquid chromatography. *Results*: COE 70 had higher polyphenol and flavonoid concentrations than COW and exhibited an excellent antioxidant effect. COE 70 suppressed UVA-induced fibroblast death by 21.3% at 25 µg/mL. It also increased *MMP-1*, *MMP-3*, *TNF-α*, and *IL-6* mRNA levels at 5–25 µg/mL compared with those in control UVA-irradiated fibroblasts. Moreover, mRNA levels of collagen type I and superoxide dismutase significantly increased, indicating the antiwrinkle and anti-inflammatory effects of the extract. Among the COE 70 components, quercitrin concentration was the highest; hence, quercitrin could be an active ingredient. *Conclusions*: COE 70 could be used as a natural antioxidant and antiwrinkle agent.

## 1. Introduction

Aging occurs following a decrease in the function of physiological factors in the human body. It can be divided into intrinsic aging, caused by genetic factors, and exogenous aging, caused by oxidative stress in the external environment. An example of exogenous aging is photoaging from exposure to sunlight [1,2,3]. Photoaging is caused by reactive oxygen species (ROS) generated when the skin is exposed to ultraviolet (UV) rays, resulting in oxidative stress and skin health deterioration [4]. ROS are a family of free radicals with one or more unpaired electrons, which can damage structural cell membranes and lipid membranes, as well as DNA. Thus, oxidative stress damages cells, leading to the onset of various diseases and skin aging [5]. Excessive ROS production is also implicated in a variety of pathological phenomena, such as mutations, cancer development, intercellular DNA degeneration, and other disorders and disease states, associated with inflammation and aging [6] including necrosis in severe cases [7]. Under oxidative stress, the epidermis becomes dry and wrinkled, capillaries enlarge, and thin collagen fibers form in the dermis, which presents morphological abnormalities of intercellular lipids [8,9,10]. In addition, excessive ROS generation and oxidative stress in the skin can cause pigmentation—spots and freckles—and various diseases, such as skin cancer [11]. Therefore, the development of antioxidants to ameliorate ROS production and maintain healthy skin has attracted research attention. For example, the mechanism underlying the anti-aging effect of carotenoids via antioxidant activity has been studied in a UVA-irradiated human skin fibroblast model [12]. Coenzyme Q10 (CoQ10) reportedly reduces DNA damage induced by UVA irradiation in human keratinocytes in vitro; it also downregulates IL-6 expression and inhibits MMP-1 production in dermal fibroblasts [13].

Recently, studies have focused on natural antioxidants that are safe and have strong effects on the skin, and extensive efforts have been made to identify functional substances in natural products [14]. Extracts from wood and plant leaves contain various antioxidant compounds, including polyphenols, flavonoids, phenolic acids, tocopherols, and tannins [15]. A variety of solvents are used to extract active ingredients from plant materials. Water and ethanol are known to easily extract phenolic derivative compounds from natural products, including substances with excellent antioxidant properties [16]. Ethanol is a natural solvent with food and cosmetic applications and is safe for human consumption. A mixture of ethanol and water is more polar than absolute ethanol, which can positively affect the total phenol yield [17]. Plant extracts, which are rich in phenolic compounds, have excellent antioxidant properties and are of interest in the pharmaceutical, food, and cosmetic industries [18]. Chamaecyparis obtusa (*C. obtuse)* is a species of Cypress (Chamaecyparis). Found in China, Japan, and Korea, it is widely cultivated for its fine wood and dark bark [19]. *C. obtuse* essential oil exerts anti-inflammatory effects through cyclooxygenase-2 (COX-2) in rats, and cypress oil *C. obtusa* reportedly has antibacterial, anti-inflammatory, and antioxidant effects [20,21,22]. Its effects against various bacteria, including *Staphylococcus aureus*, *Pseudomonas aeruginosa*, and *Escherichia coli*, have been demonstrated, and *C. obtusa* essential oil is included as an active ingredient in commercial cleaning agents and oral hygiene products [23]. A crude extract of *C. obtusa* inhibited allergic responses in an AD mouse model [24] and effectively inhibited serum levels of IgE and pro-inflammatory cytokines and mast cell overexpression in the skin [25]. It has also been found to contribute to dermatological activity [26].

Although there have been several reports on the pharmacological potential of *C. obtusa*, studies on skin cell safety and anti-aging based on its active ingredients are limited [27]. In this study, we confirmed the UVA-induced CCD-986sk cell protective effect of *C. obtusa* based on anticancer and anti-inflammatory properties. Subsequently, we investigated the effect of *C. obtusa* on skin-wrinkle-related genes. This study aimed to elucidate the overall expression of genes related to oxidation and wrinkle formation, and confirm the ability of *C. obtusa* extract to protect cells from UV rays.

## 2. Materials and Methods

### 2.1. Plant Materials

*Chamaecyparis obtusa* leaves were obtained from Mokwon Forest Engineer Co., Ltd. at Gwangyang-eup, Gwangyang-si, Republic of Korea (34.97 N, 127.58 E), in March 2020. The collection of leaves was authenticated by the plant managing director (Jin Tae Lee) of Skin Immunology Laboratory and the leaves were deposited in the herbarium of Skin Immunology Laboratory, Gyeongsangbuk-do, Republic of Korea. The collected leaves were washed, dried, and used in this study.

The water extract was obtained by boiling the leaves twice for 3 h with distilled water at 10-times the volume (*w*/*v*). To obtain the ethanol extract, 70% ethanol at 10-times (*w*/*v*) the volume was added to the leaf sample, which was then placed at room temperature (20 °C–25 °C); the sample was extracted twice over 24 h. Impurities were removed by passing the extract through a paper filter (Whatman™ Quantitative Filter Paper; Toyo Kaisha, Tokyo, Japan), which was then concentrated using a vacuum rotary evaporator (HS-10SP; Hahnshin S&T, Gimpo, Korea). Totals of 58.3 g of water-extract powder (yield of 19.45%) and 35.5 g of ethanol-extract powder (yield of 11.86%) were obtained. The final dried sample was obtained using a freeze-dryer (Freeze Dryer with Micro Concentrator MCFD; Ilshin Biobase, Gyeonggi-do, Korea). The extracts were stored at 4 °C before use, and the extraction method and yield are summarized in Appendix A.

### 2.2. Chemical and Reagents

Diphenyl-1-picrylhydrazy (DPPH), tannic acid, L-ascorbic acid, Folin Ciocalteu’s phenol reagent, 2,2’-azinobis (3-ethyl benzothiazoline-6-sulfonic acid) (ABTS), butylated hydroxyanisole (BHA), dimethyl sulfoxide (DMSO), potassium persulfate (K_2_S_2_O_8_), trizma base, 1,2,3-trihydroxybenzene, 2,3-dihydroxyphenol, pyrogallic acid, MTT, xanthine oxidase, xanthine, elastase, N-succinyl-Ala-Ala-Ala-P-nitroanilide, collagenase from *Clostridium histolyticum*, and 4-phenylazobenzyloxy-carbonyl-Pro-Leu-Gly-Pro-D-Arg were purchased from Sigma-Aldrich (St. Louis, MO, USA). Monobasic sodium phosphate and dibasic sodium phosphate were purchased from Duksan Chemical (Ansan, Korea). RIPA buffer, Dulbecco’s modified Eagle’s medium (DMEM), fetal bovine serum (FBS), and a penicillin/streptomycin mixture (P/S) were obtained from Gibco BRL Co. (Grand Island, NY, USA). Detailed information for the reagents is presented in Appendix A.

### 2.3. Antioxidant Assays

#### 2.3.1. Total Polyphenol Content

The total polyphenol content of *C. obtusa* leaf extract was measured using the Folin and Denis assay [28]. The leaf extract was dissolved in distilled water to a concentration of 1 mg/mL, and then 1 mL of the dissolved extract was added to each test tube. Folin reagent (1 mL) was subsequently added to each test tube and allowed to react for 3 min. Thereafter, 3 mL of 10% Na_2_CO_3_ was added, and the mixture was shaken vigorously for 1 h. The absorbance of the samples was measured at 725 nm, and polyphenol content was determined using a typical calibration curve (R^2^ ≥ 0.99) prepared with tannic acid (0.02, 0.04, 0.06, 0.08, and 0.1 mg/mL).

#### 2.3.2. Total Flavonoid Content

Flavonoid content in the leaves of *C. obtusa* was measured using the method described by Davies et al. [29]. Briefly, the leaf extract was dissolved in distilled water to a concentration of 1 mg/mL and added to test tubes. Thereafter, 1 mL of di-ethylene glycol reagent and 100 μL of 1 N NaOH were added to the test tubes. After vortexing, the mixture was allowed to react at 37 °C for 1 h; the absorbance of the samples was measured at 420 nm. Flavonoid content was determined using a typical calibration curve (R^2^ ≥ 0.99) prepared with quercetin (0.2, 0.4, 0.6, 0.8, and 1.0 mg/mL).

#### 2.3.3. Electron-Donating Ability

Antioxidant activity was assessed based on the scavenging activity of DPPH free radicals, according to the method described by Blois [30], with modifications. Briefly, 50 µL of 0.2 mM DPPH solution and 100 µL of extract at concentrations of 10, 100, and 1000 µg/mL were mixed with 99% ethanol in a 96-well microplate. The mixture was then incubated at room temperature for 30 min. The decolorization of DPPH was measured by the absorbance of the samples at 517 nm using an enzyme-linked immuno-sorbent assay reader (PowerWave™ XS Microplate Spectrophotometer; BioTek Instruments, Winooski, VT, USA).

#### 2.3.4. ABTS^+^ Radical Scavenging Assay

The ABTS^+^ radical scavenging assay was performed according to the method described by Brand-Williams et al. [26]. Briefly, 7 mM ABTS and 2.4 mM K_2_S_2_O_8_ were mixed and reacted in the dark at room temperature for 24 h to generate ABTS^+^ radical cations. Before use, the ABTS^+^ solution was diluted with 99% ethanol, and a spectrophotometer was used to confirm the absorbance value of 0.706 ± 0.001 at 734 nm.

#### 2.3.5. Superoxide Dismutase (SOD)-like Activity

SOD-like activity was measured using the method described by Marklund et al. [31]. Briefly, 0.2 mL of various concentrations (10, 100, and 1000 µg/mL) of the extracts, 2.6 mL of tris-HCl buffer (20 mM Tris + 10 mM EDTA, pH 8.5), and 0.2 mL of 7.2 mM pyrogallol were added to test tubes and subsequently vortexed and reacted at 25 °C for 10 min. Thereafter, 0.1 mL of 0.1 N HCl was added to stop the reaction, and the absorbance of the samples was measured at 420 nm using a UV-Vis spectrophotometer (UV-1280; Shimadzu Corp., Kyoto, Japan) to determine the effect of the extract on oxidative stress.

#### 2.3.6. Xanthine Oxidase Inhibition Assay

Xanthine oxidase inhibition activity was measured following the method described by Stirpe et al. [32]. The assay mixture comprised 0.1 mL of the sample solution (10, 100, and 1000 µg/mL), 0.6 mL of phosphate buffer (0.1 M, pH 7.5), 0.2 mL of 2 mM substrate solution, and 0.1 mL of xanthine oxidase enzyme solution (0.2 units/mL); the mixture was allowed to stand at 37 °C for 15 min. Thereafter, 0.1 mL of 1 N HCl was added to stop the xanthine oxidation reaction, and the absorbance was measured at 290 nm using a UV-Vis spectrophotometer.

#### 2.3.7. Ferric Reducing Antioxidant Power (FRAP)

The reduction ability of the sample was measured according to Oyaizu’s method [33]. Approximately 1 mL of 10, 100, and 1000 μg/mL samples and 1 mL of ferricyanide solution (1% *w*/*v*) were added to 1 mL of phosphate buffer (0.2 M, pH 6.6); the mixture was allowed to stand for 20 min before mixing at 50 °C. Thereafter, 1 mL of 10% trichloroacetic acid was added. The reaction mixture was centrifuged at 13,527× *g* for 10 min, and 1 mL of distilled water was added to 1 mL of the supernatant and 0.2 mL of 1% ferric chloride. After 10 min, the absorbance of the samples was measured at 700 nm.

### 2.4. Cell Protective Capacity

#### 2.4.1. Cell Culture and UV Irradiation

The human dermal fibroblast cell line CCD-986sk was purchased from the Korean Cell Line Bank (KCLB, Seoul, Korea). The cells were cultured in DMEM containing 10% FBS, 100 units/mL penicillin, and 100 μg/mL streptomycin in culture flasks in a CO_2_ incubator with a humidified atmosphere at 37 °C with 5% CO_2_. The UV irradiation experiment was performed as follows. CCD-986sk cells were incubated for 24 h and then stabilized by replacing the medium with serum-free DMEM. Subsequently, the medium was replaced with PBS, the cells were irradiated with UVA (10 mJ/cm^2^) for 10 s, and the extract was applied.

#### 2.4.2. Cell Viability Assay

Cell viability was determined using the MTT assay [34]. CCD-986sk cells were seeded at 1 × 10^4^ cells/well in a 96-well plate, incubated in a CO_2_ incubator for 24 h, and treated for 24 h with specific concentrations (0, 12.5, 25, 50, and 100 μg/mL) of the extract. MTT reagent (5 mg/mL) was added to the cells, which were then incubated at 37 °C for 4 h. After confirming the formation of formazan, 100 μL of DMSO was added to each well, and purple crystals were completely dissolved at room temperature for 10 min to measure the absorbance at 540 nm using an ELISA reader.

### 2.5. Anti-Aging Effect

#### 2.5.1. Elastase Inhibition Activity

The following components were added to each microtiter plate well: 50 µL of 0.4 M Tris-HCl, pH 6.8, 100 µL of various concentrations (10, 100, and 1000 µg/mL) COW and COE 70, and 50 µL of enzyme solution (ex-porcine pancreas, 0.6 unit/mL in 50 mM Tris-HCl, pH 8.6). The substrate was prepared using Cannell’s method [35]. This mixture was incubated for 30 min at 37 °C, and then 100 µL of methoxysuccinyl-L-Ala-L-Val-5-nitroanilide (1 mg/mL dissolved in dimethyl sulfoxide at 100 mg/mL, and the volume was made up with 50 mM Tris-HCl, pH 8.6) was added. The absorbance of the samples at 410 nm was immediately read using an ELISA reader and again after significant color development.

#### 2.5.2. Collagenase Inhibition Activity

Collagenase inhibition activity was measured following the method of Van Wart and Steinbrink [36], with modifications. In test tubes, 0.05 mL of various concentrations (10, 100, and 1000 µg/mL) of COW and COE 70, 0.8 mL of Tris buffer (0.05 M, pH 7.5), 0.05 mL of substrate (0.3 mg/mL 4-phenylazobenzyloxycarbon y-Pro-Leu-Gly-Pro-Arg), and 0.05 mL of collagenase (0.2 mg/mL) were added. The mixture was stirred and reacted at 25 °C for 20 min, after which 10 µL of 1 N ethyl acetate was added to stop the reaction. The collagenase reaction was measured using an ELISA reader at 345 nm.

#### 2.5.3. qPCR

The cells were seeded in a 60 mm dish at a density of 5 × 10^5^ cells/mL and incubated for 24 h; then, the medium was replaced with serum-free DMEM and the cells were stabilized. Subsequently, the medium was replaced with PBS and the cells were irradiated with UVA (10 mJ/cm^2^). The cells were cultured for 24 h in a medium containing the extract (5, 12.5, and 25 µg/mL) and the positive control 25 µM epigallocatechin *gallate* (EGCG). After incubation, the supernatant was removed and washed twice with Hank’s Balanced Salt Solution (Sigma Chemical Co., St. Louis, MO, USA). The cell wall was then broken using TRI-Solution (Invitrogen, Carlsbad, CA, USA), chloroform was added to the aqueous solution layer and organic solvent layer, and the phenol component was removed by air drying. RNase-free deionized water was added, and the pellets were dissolved to obtain total RNA. RNA was quantified using Nanodrop (Microdigital, Seongnam, Korea), and cDNA was synthesized using the Promega GoScript™ Reverse Transcription System (Thermo Fisher Scientific, Waltham, MA, USA) and amplified using TB Green^®^ Premix Ex Taq II (Takara Bio, Otsu, Japan), per the manufacturers’ instructions. Detailed information of reagents and kits is presented in Appendix A. The PCR cycles were repeated 40 times for denaturation at 94 °C, annealing at 55 °C for 30 s, and extension at 72 °C for 30 s. The primer pairs are shown in Table 1.

### 2.6. HPLC Analysis

The sample solution was obtained by dissolving 1 mg of COE 70 in 1 mL of methanol solution. Thereafter, the supernatant was filtered through a 0.22 μm syringe filter. The HPLC system used for the analysis was a Hitachi Lachrom Elite^®^ HPLC system (Hitachi Instruments Inc., Danbury, CT, USA) equipped with an autosampler and a UV detector. Chromatographic separation was accomplished using a Zorbax Eclipse XDB-C18 (250 mm × 4.6 mm, 5 μm) analytical column. The HPLC conditions are summarized in Appendix A.

### 2.7. Statistical Analysis

The Statistical Processing of Experimental Results Social Sciences (SPSS) software package (Version 22.0; IBM Corp., Armonk, NY, USA) was used to analyze the data, which are presented as mean and standard deviation (SD) for each treatment group. The significance of differences was analyzed using the analysis of variance (ANOVA). The results were analyzed using Duncan’s multiple test and a *t*-test. Statistical significance was set at *p* < 0.05.

## 3. Results and Discussion

### 3.1. Antioxidant Effect of C. obtusa Extract

Natural and synthetic antioxidants reduce the oxidation of DPPH radicals. The reduction of DPPH in alcoholic solutions in the presence of hydrogen-donating antioxidants results in DPPH-H due to the formation of non-radicals. The DPPH experiment is an easy and quick way to assess the antioxidant activity of a component [37,38]. Here, in the evaluation of DPPH radical scavenging ability, COE 70 at 1000 g/mL had the highest concentration of ascorbic acid (99.6%); at this concentration, COE 70 and COW showed efficacies of 92.1 and 68.6%, respectively (Figure 1a). The ABTS radical scavenging assay was used to measure antioxidant activity using the principle that the blue-green ABTS radical cation formed by the reaction with K_2_S_2_O_8_ is removed by the antioxidant substance in the extract and decolorized [39].

As the ABTS radical scavenging assay can measure the scavenging activity of both polar and non-polar samples, the application range of the assay is wider than that of the DPPH radical scavenging assay [40]. In addition, the DPPH radical is a free radical, whereas the ABTS radical is a cation radical, both of which have different substrate characteristics, and the degree of binding between free radicals and cation radicals may be different depending on the characteristics of the extract [41]. It is necessary to analyze both types of radical scavenging ability. In the evaluation of ABTS radical scavenging ability, the two extracts at 1000 µg/mL had the highest concentration of ascorbic acid (90.8%); at this concentration, COE 70 and COW showed efficacies of 73.7 and 54.7, respectively (Figure 1b). Xanthine oxidase acts as a rate-limiting enzyme in the terminal oxidation of all purines, leading to their oxidation by superoxide radicals or hydrogen peroxide. Superoxide radical inhibition by xanthine oxidase is related to the scavenging action of superoxide radicals and inhibition of xanthine oxidase, and this process has biological significance as it suppresses the generation of free radicals [42]. The positive control group, ascorbic acid, showed 91.0% inhibitory activity at a concentration of 1000 µg/mL; COE 70 showed 76.5% and COW showed 45.0% xanthine oxidase inhibitory activity, indicating that the 70% ethanol extract has good efficacy (Figure 1c).

SOD is involved in the elimination of O_2_ (superoxide) in the body, and the active oxygen generated in the body causes oxidative disorders. Through its activity, SOD controls the levels of several ROS and reactive nitrogen species, thereby regulating various aspects of the cell cycle involving signaling functions [35,43]. To investigate the antioxidant activity of superoxides, we measured superoxides and the auto-oxidation reaction of pyrogallol, which reacts to produce a brown substance. The results were similar to those of antioxidant activity measured by DPPH, with COE 70 showing better activity than COW (Figure 1d).

The FRAP assay is used to measure the total antioxidant power of a sample by analyzing the process of reducing ferric ions to ferrous ions by a colored ferrous-TPTZ complex. Based on the fact that most antioxidants have reducing power, the assay is based on the principle that the ferric-TPTZ (Fe^3+^-TPTZ) complex is reduced to ferrous-TPTZ (Fe^2+^-TPTZ) by the reducing agent at low pH [44,45]. In this study, the concentrations of FRAP in COE 70, COW, and ascorbic acid at 1000 μg/mL were 66.8, 39.1, and 87.0 μg/mL, respectively (Figure 1e). Using activity analysis according to the ratio of hot water and ethanol of cypress leaves, a previous study confirmed that the polyphenol content, antioxidant activity, and antibacterial activity increased when the ethanol content was 40% or more compared with the hot water extract of cypress leaves. In this study, when extracted with 70% ethanol, the antioxidant activity was higher than that of the hot water extract, confirming the efficacy of the ethanol extract [46].

According to some studies [47], the antioxidant activity of natural products is correlated with polyphenol content; this is consistent with our findings (Table 2). The total phenolic content was calculated from the gallic acid (GA) standard curve, and the total phenolic content was calculated from the catechin standard curve using a regression equation. The COE 70 samples had higher polyphenol and flavonoid concentrations than the COW samples. Furthermore, the *C. obtusa* extract showed excellent efficacy in the antioxidant experiments, indicating strong free radical scavenging activity.

### 3.2. Effect of C. obtusa Extract on Fibroblast Damage under UVA Irradiation

Oxidative stress plays an important role in the induction of photoaging [48]. In addition, oxidative stress regulates the synthetic activity of dermal fibroblasts, thereby reducing the production of elastin, collagen, and other basic compounds, such as glycosaminoglycans and proteoglycans, by either inducing photoaging or increasing the rate of collagen degradation [49]. A cell viability analysis was performed to determine the cytotoxicity of COE 70. CCD-986 fibroblasts were treated with 0, 12.5, 25, 50, or 100 µg/mL COE 70 for 24 h, and cell survival was determined using MTT reagents (Figure 2a). The photoprotective ability of COE 70 in CCD-986sk cells irradiated with UVA (10 mJ/cm^2^) was evaluated using the MTT assay. The cell survival rate increased in the group treated simultaneously with UVA and COE 70 (at concentrations of 12.5 and 25 µg/mL, respectively) compared with that in the group treated only with UVA, confirming that COE 70 possesses photoprotective abilities (Figure 2b).

### 3.3. Anti-Aging Effect of C. obtusa Extract

Antioxidant treatment reduces the negative effects of free radicals and age-related diseases. It is also known to be a useful method to protect the skin from aging [50]. ROS induce the secretion of MMPs, elastase, and collagenase enzymes, causing dry skin and inflammatory wrinkles [51]. In dermal tissue, collagen and elastin are associated with the elasticity of the skin and form a network structure. As elastin is decomposed by elastase, which breaks the bond of the skin network structure, elastase is known to be the main cause of wrinkling. Elastase inhibitors, such as ursolic acid, contribute to an improvement in skin wrinkles [52]. In addition, as one of the enzymes of leukocyte granulation that decomposes elastin in the body, elastase activity is extremely high in abnormal tissues, which directly causes tissue destruction, wrinkling, and loss of elasticity in the skin [53]. Phenolic and flavonoid content has been found to be associated with antioxidant activity, and polyphenols have been reported to have substantial anti-aging effects. In addition, phenolic compounds are known to have collagenase, elastase, and hyaluronidase inhibitory activities [54]. To evaluate the effect of *C. obtusa* on skin wrinkle improvement, collagenase and elastase enzyme inhibitory activities were measured in this study. COE 70 and the positive control ascorbic acid, at 1000 µg/mL, showed similar collagenase and elastase inhibition rates, confirming that *C. obtusa* is effective in wrinkle improvement (Figure 3a,b).

The mRNA levels of *MMP-1*, *MMP-3*, *TNF-α*, *IL-6*, collagen type I, and *SOD 1* in stimulated fibroblasts were investigated using qPCR. At a concentration of 25 µg/mL, COE 70 showed excellent efficacy by reducing the mRNA levels of MMP-1, MMP-3, TNF-a, and IL-6 by 39.2%, 72.4%, 79.6%, and 87.5%, respectively, when compared with fibroblasts irradiated with UVA alone. Compared with the IL-6 and TNF-a cytokine expression of Nakai Extract, a natural product with excellent antiwrinkle and anti-inflammatory efficacy, COE 70 had higher efficacy [55].

MMP-1 mRNA expression in TNF-α-induced cells of *Cosmos caudatus* Kunth showed similar efficacy to COE 70 at a concentration of 25 µg/mL, and MMP-3 showed superior efficacy in COE 70 [56]. The mRNA levels of collagen type I and *SOD 1* significantly increased as the concentration of the extract increased (Figure 3c–h). In a previous study, quercitrin increased collagen levels and decreased MMP-1, IL-6, and ROS levels. Quercitrin has also been shown to downregulate IL-6 expression, contributing to a reduction in the inflammatory response [57]. In addition, amentoflavone has been shown to inhibit DNA damage and MMP-1 protein expression in UVB-irradiated HaCaT cells [58]. In our study, the results for COE 70 were similar to those of the previous studies, indicating a high content of quercitrin among the active ingredients of COE 70. The moderate phenolic and flavonoid concentrations in plants might be related to their antioxidant activity and anti-aging effects.

### 3.4. Quantitative Analysis of Four Structural Compounds in C. obtusa

We quantitatively analyzed quercitrin, myricetin, amentoflavone, and hinokiflavone in *C. obtusa* ethanol extract. HPLC chromatograms for COE 70 are presented in Figure 4a, and the chemical structures of the standard compounds are presented in Figure 4b. The concentrations of the above compounds ranged from 0.26 to 12.4 mg/g. Table 3 presents the results of the quantitative analysis. Among the four compounds, quercitrin (12.4 mg/g) was the most abundant.

## 4. Conclusions

In this study, we evaluated the antioxidant and antiwrinkle efficacies of *C. obtusa* extract in UVA-irradiated CCD-986sk cells. COE 70 markedly promoted the synthesis of collagen type I and inhibited the mRNA expression of *MMP-1* and *MMP-3*, as well as the production of the pro-inflammatory mediators IL-6 and TNF-α, in CCD-986sk cells following UVA irradiation. In addition, COE 70 increased the mRNA expression of SOD 1, an enzyme that controls the levels of ROS. Using HPLC, we identified quercitrin, myricetin, amentoflavone, and hinokiflavone as the functional components of COE 70. Among these four compounds, quercitrin was the most abundant, and it was confirmed that it acts as an active ingredient involved in the biological efficacy and antiwrinkle activity of COE 70. In the future, we aim to identify additional active ingredients in COE 70 and conduct diverse studies in artificial skin models, including skin cells, to determine whether these ingredients exhibit antiwrinkle activity. The results of this study elucidate the potential therapeutic application of COE 70 for aging and wrinkling.

## Figures and Tables

**Figure 1 medicina-59-00755-f001:**
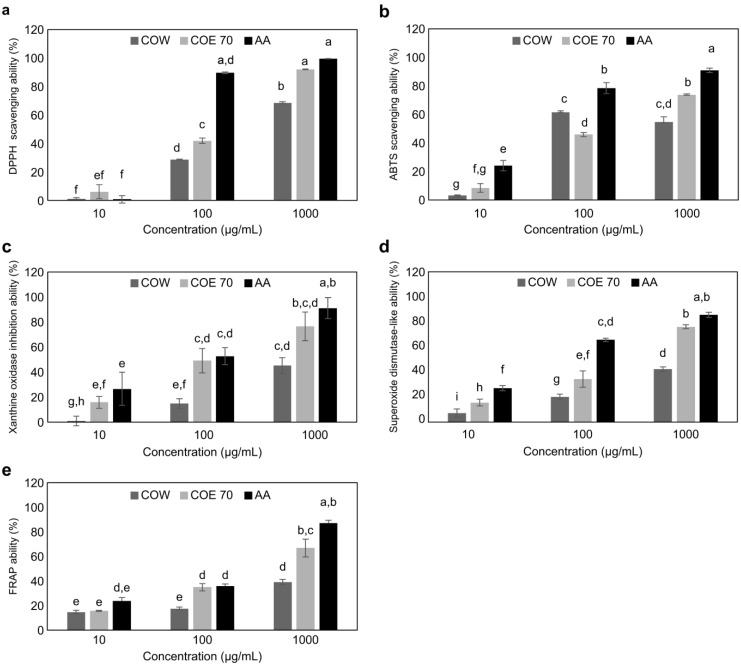
Antioxidant effects of *C. obtusa* leaf extract. Antioxidant activity was evaluated by (**a**) electron-donating ability, (**b**) ABTS⁺ radical scavenging assay, (**c**) SOD-like activity, (**d**) xanthine oxidase assay, and (**e**) FRAP. AA: ascorbic acid. The values are expressed as mean ± SD (*n* = 3). All experiments were independently conducted three times. Values with different characters (a–g) are significantly different from each other, as determined by ANOVA and Duncan’s multiple range test (*p* < 0.05).

**Figure 2 medicina-59-00755-f002:**
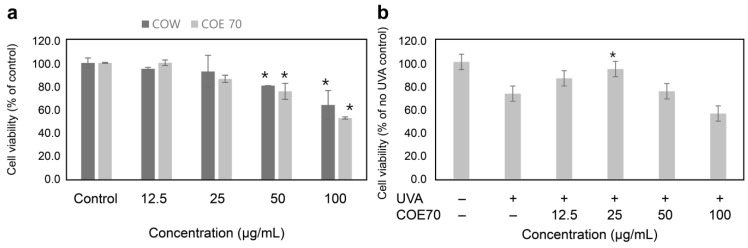
Viability of CCD-986sk fibroblasts treated with *C. obtusa* leaf extract. (**a**) The cells were cultured in 96-well plates and treated with the final extract at concentrations of 0, 12.5, 25, 50, and 100 µg/mL for 24 h. Survival rates of cells treated with the extracts were calculated using the MTT assay. (**b**) The cells were irradiated with UVA (10 mJ/cm^2^) in the absence or presence of *C. obtusa* extract in the medium. All experiments were independently conducted three times. Statistical analysis was performed using ANOVA and the results were compared with those of the experimental controls with Bonferroni’s test; * *p* < 0.05 compared to the control (**a**) and UVA (10 mJ/cm^2^)-irradiated (**b**) groups.

**Figure 3 medicina-59-00755-f003:**
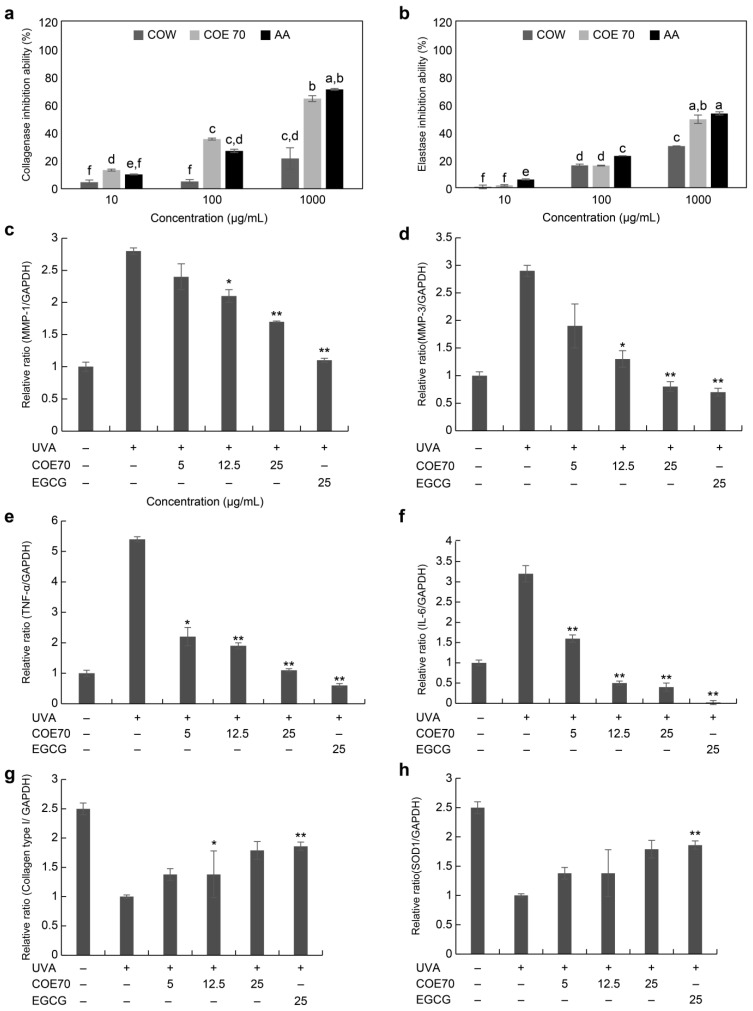
Anti-aging effect of *C. obtusa* leaf extract. Antioxidant activity was measured by (**a**) elastase inhibition activity and (**b**) collagenase inhibition activity. The mRNA level of (**c**) *MMP-1*, (**d**) *MMP-3*, (**e**) *TNF-α*, (**f**) *IL-6*, (**g**) collagen type I, and (**h**) *SOD 1* in UVA-irradiated fibroblasts using qRT-PCR. Each value is expressed as the mean ± SD (n = 3). All experiments were independently conducted three times. Values with different characters (a–h) are significantly different from each other, as determined using ANOVA and Duncan’s multiple range test (*p* < 0.05). Nor: non-UVA-irradiated cells; Con: UVA-irradiated cells without *C. obtusa* treatment (**c**–**h**). Statistical analysis was performed using ANOVA and the results were compared with those of the experimental controls with Bonferroni’s test; * *p* < 0.05 and ** *p* < 0.01 compared to the control (**a**) and UVA (10 mJ/cm^2^)-irradiated (**b**) groups.

**Figure 4 medicina-59-00755-f004:**
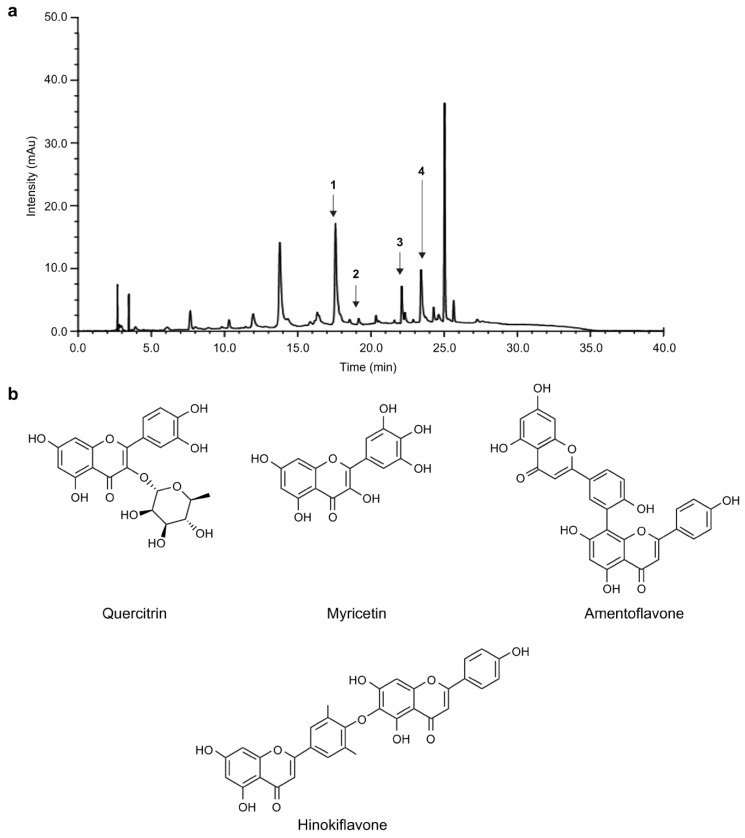
High-performance liquid chromatography. Chromatogram of the ethanol extract of (**a**) *C. obtusa* and (**b**) the quercitrin, myricetin, amentoflavone, and hinokiflavone standard compounds.

**Table 1 medicina-59-00755-t001:** Sequences of primers used in the PCR analysis of CCD-986sk fibroblasts.

Gene	Primer	Sequence (5′-3′)	Expected Size (bp)	Accession Number
COL1A1	Forward Reverse	GATTCCCTGGACCTAAAGGTGC AGCCTCTCCATCTTTGCCAGCA	107	NM_000088.4
MMP-1	Forward Reverse	GGGCTTGAAGCTGCTTACGA ACAGCCCAGTACTTATTCCCTTTG	74	NM_002421.4
MMP-3	Forward Reverse	GATGCCCACTTTGATGATGATGAA AGTGTTGGCTGAGTGAAAGAGACC	119	NM_002422.5
TNF-α	Forward Reverse	CCCAGGGACCTCTCTCTAATC GGTTTGCTACAACATGGGCTACA	97	NM_000594.4
IL-6	Forward Reverse	ACTCACCTCTTCAGAACGAATTG CCATCTTTGGAAGGTTCAGGTTG	149	NM_000600.5
SOD1	Forward Reverse	ACTGGTGGTCCATGAAAAAGC AACGACTTCCAGCGTTTCCT	83	NM_000454.5
GAPDH	Forward Reverse	ACTGCTTAGCACCCCTGGCCA TTGGCAGTGGGGACACGGAAG	253	NM_001357943.2

**Table 2 medicina-59-00755-t002:** Total phenolic and flavonoid content of *C. obtusa*.

Extract	Total Phenolic Content (mg TAE/g)	Total Flavonoid Content (mg QE/g)
Water	45.04 ± 1.07	0.99 ± 0.05
70% Ethanol	103.41 ± 5.25	1.77 ± 0.08

TAE, tannic acid equivalents. QE, quercetin equivalent. Values are means ± standard deviation (n = 3).

**Table 3 medicina-59-00755-t003:** Regression equation, linearity, limits of detection, limits of quantification, and content of the four standard compounds.

Compound	Linear Range (mg/L)	Response Slope (a)	Response Factor (b)	Correlation Coefficient (R^2^)	LOD (mg/L)	LOQ (mg/L)	Content (mg/g)
Quercitrin	5–100	20,574	−63,278	0.9913	0.641	1.944	12.4
Myricetin	10–200	0.364	−5.115	0.9958	16.437	49.808	0.26
Amentoflavone	5–100	41,878	−27,887	0.9998	0.848	2.569	5
Hinokiflavone	5–100	0.4401	−0.4674	0.9795	18.774	56.891	0.88

## Data Availability

The data presented in this study and additional information on materials and methods are available on request from the corresponding author.

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
