# Peer review of "Biological Activity and Component Analyses of Chamaecyparis obtusa Leaf Extract: Evaluation of Antiwrinkle and Cell Protection Effects in UVA-Irradiated Cells"

_medicina, 2023, doi:10.3390/medicina59040755_

Round 1
Reviewer 1 Report
This paper titled “Biological activity and component analyses of Chamaecyparis obtusa leaf extract: Evaluation of antiwrinkle and cell-protection effects in UVA-irradiated cells” has studied the antioxidant, cell protection from UVA irradiation, and anti-aging properties of C.Obtusa leaf 70% ethanol extract and it’s phytochemical profile.
Major comment
1) The authors isolated 4 compounds from chamaecyparis obtusa. I would recommend that 4 isolated compounds need skin protection assay against UVA exposure.
2) The author should revise the whole manuscript and write more crisp and concise to avoid redundant and lengthy sentences.
Minor comments
Please kindly fix the following typo errors in the manuscript
1. Section 3.2 line 324, The concentration 100 ug/mL is missing, typo error 70% ug/mL,
2. Fig 2A, the Y axis naming error “control of %” or is it “% of control?” and in 2B is it “ % of UVA” or no UVA control ?
Reviewer 2 Report
The article written by Young-Ah Jang et al. evaluated the antioxidant activity and antiwrinkle effect of C. obtusa extract to determine its potential as a cosmetic material However, while the purpose of doing such work is important to identify anti-aging abilities of natural plant extracts, i am worried about the novelty of the article.When doing research in general, I don't see it novel or influental enough as a separate publication. If the editor sees that the manuscript should be published, some major and minor issues must be taken care of before that.
Major comments
1. Generally,methods and results cannot simply be stated at the end of the the part of introduction and need to be modified. Besides, why the authors use 70% ethanol to extract C. obtusa? If there any other studies using different concerntration of ethanol to extract, please cite relevant references.
2. I noticed that a research work published in 2022(Kwon YJ et al. PMID: 34364971)suggested the use of C. obtusa extracts(99% ethanol) as therapeutic approach for inflammatory diseases. Did the authors detected the antioxidant effects between 70% and 99% ethanol extracts of C. obtusa?
3. What are the antioxidant advantages of COE 70% compared with other plant extracts? Please state.
4. The authors evaluated the antioxidant and antiwrinkling efficacies of C. obtusa extract in UVA-irradiated CCD-986sk cells.The related mechanism and signal pathway showed be detected.
Minor comments
1. In line362-363 ,The authors found that COE 70% increased the mRNA levels of MMP-1, MMP-3, TNF-α, and IL-6 in UVA-irradiated fibroblasts at concentrations of 5, 10, and 20 µg/mL.But in figure 3, the labels of concentration in the diagram are5, 12.5, and 25 µg/mL.
2. The list of abbreviations showed be used, I would expect a complete list of abbreviations being used in the text
3. The authors showed add the limitation part of the manuscript.
